# The structure and predictive value of intrinsic capacity in a longitudinal study of ageing

John R Beard ![ORCID] ,[1,2] A T Jotheeswaran,[1] Matteo Cesari,[3] Islene Araujo de Carvalho ![ORCID] [1]

[1]Department of Ageing and Life Course, World Health Organization, Geneve, Switzerland
[2]University of New South Wales, Sydney, Australia
[3]Fondazione IRCCS Ca' Granda Ospedale Maggiore Policlinico, Università di Milano, Milano, Italy

**Correspondence to**
Dr. Islene Araujo de Carvalho;
araujodecarvalho@who.int

## ABSTRACT

**Objectives** To assess the validity of the WHO concept of intrinsic capacity in a longitudinal study of ageing; to identify whether this overall measure disaggregated into biologically plausible and clinically useful subdomains; and to assess whether total capacity predicted subsequent care dependence.

**Design** Structural equation modelling of biomarkers and self-reported measures in the English Longitudinal Study of Ageing including exploratory factor analysis, exploratory bi-factor analysis and confirmatory factor analysis. Longitudinal mediation and moderation analysis of incident care dependence.

**Settings** Community, United Kingdom.

**Participants** 2560 eligible participants aged over 60 years.

**Main outcome measures** Activities of daily living (ADL) and instrumental activities of daily living (IADL).

**Results** One general factor (intrinsic capacity) and five subfactors emerged: locomotor, cognitive; psychological; sensory; and 'vitality'. This structure is consistent with biological theory and the model had a good fit for the data ($\chi^2$=71.2 (df=39)). The summary score of intrinsic capacity and specific subfactors showed good construct validity. In a causal path model examining incident loss of ADL and IADL, intrinsic capacity had a direct relationship with the outcome—root mean square error of approximation (*RMSEA*=0.02 (*90% CI 0.001 to 0.05*) and RMSEA=0.008 (*90% CI0.001 to 0.03*) respectively—and was a strong mediator for the effect of age, sex, wealth and education. Multimorbidity had an independent direct relationship with incident loss of ADLs but not IADLs, and also operated through intrinsic capacity. More of the indirect effect of personal characteristics on incident loss of ADLs and IADLs was mediated by intrinsic capacity than multimorbidity.

**Conclusions** The WHO construct of intrinsic capacity appears to provide valuable predictive information on an individual's subsequent functioning, even after accounting for the number of multimorbidities. The proposed general factor and subdomain structure may contribute to a transformative paradigm for future research and clinical practice.

## INTRODUCTION

In 2015, the WHO released the *World report on ageing and health*, which proposed a public health framework for action on population

### Strengths and limitations of this study

► To our knowledge this is the first large population-based longitudinal analysis to examine the structure and predictive validity of the WHO concept of intrinsic capacity. We applied a rigorous psychometric approach for constructing a valid measurement model using commonly measured biomarkers and self-reported measures, allowing us to create a theoretically error-free composite score for intrinsic capacity, which was used in all analysis.

► We used longitudinal data to minimize the potential for reverse causality and adjusted for multimorbidity to minimise confounding-by-disease; however, the potential of residual confounding cannot be completely eliminated.

► This study shows that many of the commonly used assessments of health and functioning in older age have common variance (ie, they are possibly measuring one underlying trait of an individual's health status) that is consistent with the WHO concept of intrinsic capacity.

► This composite measure was structured in a way that is consistent with biological theory.

► However, it is important to note that the measures included in the English Longitudinal Study on Ageing study are neither complete nor random. They were chosen to inform specific research questions of interest to the investigators, rather than to create an overall measure of intrinsic capacity. The consideration of other variables might influence both the overall score and the subfactor structure.

ageing.[1 2] Central to the Report is a new conceptual model for '*Healthy Ageing*'. Rather than considering healthy ageing from the perspective of the presence or absence of disease, this functioning-based approach is oriented around building and maintaining the ability of older people to be, and to do, the things they have reason to value. The Report proposes that this 'functional ability' is determined by the 'intrinsic capacity' of the individual, the environments in which they live and the interaction between the individual and these environments. However, while the

Report considers intrinsic capacity to be 'all the physical and mental capacities' that an individual can draw on at any point in time, it does not provide a detailed description of the components of capacity, how they might be structured or how capacity and its components may be measured and monitored.

This reframing of the concept of healthy ageing builds on a growing body of research exploring patterns and determinants of functional status in older people. Many of these studies examine functioning in areas such as physical performance or cognition,[3] [4] and increasingly they are applying a life course perspective.[5] At the same time there is growing interest in the biological underpinnings of ageing and in identifying ways to measure 'biological' age as distinct from chronological age.[6] This work all serves to better capture the heterogeneity that is a hallmark of ageing and helps researchers and clinicians advance from stereotypical notions of older age, and towards more personalised interventions to foster healthy ageing.

There has also been significant work identifying measures that might assess different domains of functioning at different stages in life.[7] However, there is less research and less agreement on how functional approaches for specific domains might together reflect the *overall* health status of older individuals.[8] [9] It also remains unclear how specific functional domains such as locomotor and cognitive capacity relate to each other, and how the deficits in the complex and dynamic biological systems that underpin ageing relate to these more overt expressions of an individual's capacity.[10]

Broad self-reported measures of health and well-being such as the Short form 36 (SF36) and General Health Questionnaire (GHQ) attempt to capture this heterogeneity, but do not consider key capacities (for example cognitive capacity), and can have difficulty distinguishing between the contribution of individual or environmental level factors to functional status.

Distinguishing between capacity and ability is also a problem for other commonly used measures of overall functioning in older age including Instrumental Activities of Daily Living (IADLs) or Activities of Daily Living (ADLs). Losses of IADLs and ADLs are also generally only observed with very significant decrements of functioning,[11] while the WHO model suggests that changes in capacity are likely to start much earlier in life. Understanding the factors that influence levels and trajectories of overall capacity in relatively robust people before they experience these significant losses may help identify interventions earlier in the life course, and could be useful in self-care and clinical practice. Broad based outcomes like this could be useful in other ways too: for example as a way of comparing the relative benefits of interventions on different functional domains or in different organ systems.

Continuous measures of intrinsic capacity that are sensitive to subtle changes and that distinguish between the individual and their context would thus enable a much better understanding of functioning at both a population and individual level. However, this would first require a clearer conceptualisation of the intrinsic capacity construct.

To progress work in this area, we examined data from the English Longitudinal Study on Ageing (ELSA) to assess whether a range of commonly collected biomarkers and self-reported measures might provide a useful estimate of intrinsic capacity, and whether this construct predicted subsequent outcomes in relatively robust older people after accounting for the number of health conditions a participant may be experiencing. We examined the factor structure of the total capacity score to identify relevant sub factors and used structural equation modelling to assess longitudinally the direct and indirect relationships of the total intrinsic capacity score, personal characteristics and multimorbidity with subsequent IADL or ADL loss.

## METHODS
### Study description
ELSA is an ongoing study of a nationally representative sample of the English population aged ≥50 years[12]. Participants were recruited from households that were included in the Health Survey for England in 1998, 1999 and 2001, and then followed up every 2 years with detailed health examinations through nurse visits taking place every 4 years. Data were collected via face-to-face assessments using computer-assisted personal interviews and a self-completion questionnaire. In addition, a trained nurse visited participants in waves two, four and six to measure physical functioning and collected the blood samples which were then analysed to generate biomarker data. In ELSA the response rates varied across the waves with 67% in wave 1, 82% in wave 2, 73% in wave 3, 74% in wave 4% and 80% in wave 5.[12] The inclusion criteria for the present study include (a) participants aged over 60 years included in the nurse visit, (b) consent to provide blood sample, (c) no missing data on main exposure (intrinsic capacity) indicators, and (d) follow-up outcome data available in wave 5 (2010/2011). Applying these criteria led to a total study sample of 2352 participants (figure 1).

### Patient involvement
All participants were required to provide informed written consent. All ELSA data are anonymous and freely accessible from the UK Data Service Discover. Only data contained within the ELSA database were included in the analyses. No patients were involved in the development of the research question, study design or interpretation of the data in this study.

### Measures
#### Intrinsic capacity
We considered measures collected in ELSA that might provide objective estimates of aspects of intrinsic capacity based on the following criteria: (a) prior evidence

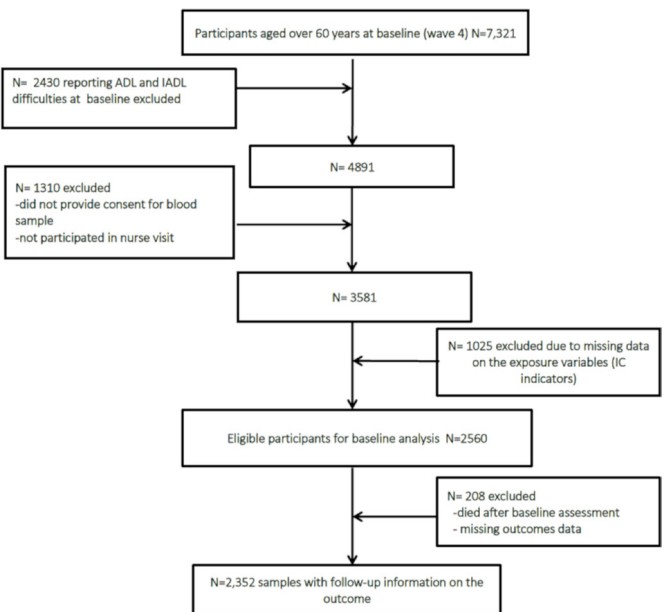

Participants aged over 60 years at baseline (wave 4) N=7,321

N= 2430 reporting ADL and IADL difficulties at baseline excluded

N= 4891

N=1310 excluded
-did not provide consent for blood sample
-not participated in nurse visit

N= 3581

N= 1025 excluded due to missing data on the exposure variables (IC indicators)

Eligible participants for baseline analysis N=2560

N= 208 excluded
-died after baseline assessment
- missing outcomes data

N=2,352 samples with follow-up information on the outcome

**Figure 1** Flow of study members into the analytical sample: the English Longitudinal Study of Ageing. ADL, activities of daily living ; IADL, instrumental activities of daily living; IC, intrinsic capacity.

supporting an association with at least one aspect of capacity, and (b) ability to distinguish between high and low physical or mental capacity at older ages and sensitivity to detect change within and between individuals over time.

Walking speed: Each participant aged 60 and above was eligible for the *timed walk* test. In addition, prior to the actual test, respondents were asked if they had any problems from recent surgery, injury, or other health conditions that might prevent them from walking. Only persons aged at least 60 years, willing to do the test, and able to walk (walking aids were permitted) were asked to walk 8 feet (2.4m) at their usual walking pace, twice.[13] The time for both walks was recorded separately. In our analysis we use the mean speed (measured in m/s) of the two trials.

Chair-stand test: The chair stand test, a measure of physical performance, assessed the time required to rise from a chair to a full standing position five times with arms folded across the chest, with slower times reflecting worse function.[14] The test incorporated the use of respondent's own armless, straight backed chair. The time taken for full stand was recorded in seconds. Respondents were considered as ineligible if they could not stand up without assistance; the use of walking aids, such as a walker or cane, was not permitted. The test was stopped if the respondent became too tired or short of breath; if the participant used their hands; if after 1 min, the participant had not completed all the rises; or if the nurse felt concerned for the respondent's safety.

Balance: Static balance was evaluated in three separate and progressively more difficult tests which formed part of the Short Physical Performance Battery.[15] Participants

were ineligible for the tests if they were chair-bound or wheelchair-based; if it became clear after discussion that they were too unsteady on their feet; if they found it painful to stand; or if either the nurse or the participant considered the test unsafe. We used three components of the balance test (an additional two components were performed by younger participants only): side-by-side, semi-tandem and full tandem. (A) Side-by-side stand: Participants were asked to stand with feet together, side-by-side, for at least 10s, using their arms, bending their knees or moving their body to maintain balance, but not moving their feet. If the participant was unable to hold the position for 10s, a score of zero was recorded and no further tests attempted. Those able to hold the position for 10s moved on to the semi-tandem stand. (B) Semi-tandem stand: Participants had to stand with the side of the heel of one foot touching the big toe of the other foot for at least 10s. Participants unable to hold the position for 10s scored one and no further tests were attempted. Those able to hold the position for 10s moved on to the full-tandem stand. (C) Full-tandem stand: For this test, participants had to stand with the heel of one foot in front of and touching the toes of the other foot. Those unable to hold this position for at least 3s scored no additional points; those able to hold the position for at least 3 but less than 10s scored one point for this test; and those able to hold the position 10s or longer scored two points for this test. The maximum possible score from all three tests was four points: one point each from the side-by-side and semi-tandem tests, and two points from the full-tandem test.

Grip strength: The grip strength test is a test for upper body strength.[16] Handgrip strength (kg) of the dominant hand was assessed using a handheld dynamometer, with the average (mean) of three measures used in the analyses. Three values were recorded for each hand, starting with the non-dominant hand and alternating between hands. Any measurements carried out incorrectly or participants refused to perform the test were not included.

Forced expiratory volume (FEV): Lung function was measured using a NDD Easy On Spirometer.[17] Willing and eligible respondents were asked to stand or seated, take a deep breath and blow into the spirometer as hard as they could. Respondents were then required to repeat the procedure to give three technically satisfactory blows. The highest technically satisfactory measure of FEV in 1 s (FEV1) was used in the analysis. The protocol required three successful measurements to be completed. An unsatisfactory blow included any of the following: an unsatisfactory start with excessive hesitation; laughing or coughing, especially during the first second; a Valsalva manoeuvre; leakage of air around the mouthpiece; obstruction of the mouthpiece by tongue or teeth; obstruction of the spirometer flow head outlet by hands.

Blood assay: A trained nurse collected biomarker data from all participants not meeting exclusion criteria. Viable blood samples were obtained from 6188 respondents (75.6% of wave 4 participants). Detailed information

on the technicalities of the blood analysis, the internal quality control and the external quality assessment for the laboratory have been described elsewhere.[18] Dehydroepiandrosterone (DHEA(S)) levels from serum was performed using the Roche DHEA(S) assay that is a competitive immunoassay using electrochemiluminescence technology (analytical range: 0.003–27 μmol/L).[19] Haemoglobin level (g/L) was measured with two Abbott Diagnostics Cell-Dyn 4000 analysers.[20] Insulin-like growth factor 1 (IGF-1) values are reported as whole numbers (range: 3–200 nmol l–1).[21]

Sensory: Hearing and vision impairments were measured using self-reported,[22 23] validated questions previously demonstrated to be accurate when compared with objective measures. Hearing status was assessed by asking participants to rate their hearing (using a hearing aid if they used one) as excellent, very good, good, fair, or poor. For vision, participants were also asked 'How good is your eyesight for seeing things at a distance, like recognising a friend across the street' and 'How good is your eyesight for seeing things up close, like reading ordinary newspaper print'. Response options (excellent/very good/good/fair–poor) were categorised as above.

Cognitive: The ELSA data include scores on three tests of cognitive function: verbal fluency, delayed verbal memory and attention.[24] Verbal (semantic) fluency was assessed by asking participants to name as many animals as they could think of in 1 min. Delayed verbal memory was assessed using lists of nouns presented aurally. Attention was assessed using a letter cancellation task. Scores on these tests were used as measures of three kinds of cognitive function: the scores on the animal naming task were taken as a measure of executive function,[25] the sum of the scores on the delayed recall tasks were taken as a measure of memory, and the scores on the letter cancellation task were taken as a measure of processing speed.[26]

Affect: Affect was measured using the eight-item Center for Epidemiological Studies-Depression (CES-D) scale.[27] Five of the eight CES-D items (ie, felt depressed, was happy, felt lonely, enjoyed life, felt sad) were depressed mood items, while the remaining three (ie, everything was an effort, restless sleep and could not get going) were somatic complaints items. We derived a summary CES-D score by adding responses to all eight dichotomous questions (possible range: 0–8).

Sleep: To assess sleep disturbance, participants were asked about the frequency of delay in falling asleep, inability to stay asleep, waking up tired and disturbed sleep in the previous month.[28] Response categories were no difficulties, less than once a week, once or twice a week and three times or more a week. These response codes were given a numerical score (1 to 4) and then items were summed and a total score created. The total score ranged between 4 and 16, and showed a normal distribution, with a mean score of 8.8 (SD 3.2).

## Other covariates

We also extracted data on other sociodemographic and medical covariates, recorded at wave 4, that may potentially confound the associations between intrinsic capacity and care dependence. These included chronological age, sex, education (no education, intermediate and higher education), total non-pension net wealth in quintile as a proxy measurement of socioeconomic status and multimorbidity (self-reported information on doctor diagnosed diabetes, hypertension, stroke, heart diseases (myocardial infarction, congestive heart failure, angina), chronic obstructive pulmonary disease, asthma, arthritis, osteoporosis, cancer, Parkinson's disease, Alzheimer's disease and other dementia.[29]

## Measures of outcome

*Care dependence:* The outcome of interest for longitudinal analysis—incident care dependence—was chosen because it was an overall measure of functioning that was assessed independently from the functional characteristics included in the intrinsic capacity construct. Care dependence was assessed using self-reported limitations in the Basic ADL and IADL.[30] Respondents were asked to exclude any difficulties expected to last less than 3 months. ADL included six activities: dressing, walking across a room, bathing or showering, eating, getting in or out of bed, using the toilet. IADL included seven activities: using a map to get around in a strange place, preparing a hot meal, shopping for groceries, making telephone calls, taking medications, doing work around the house or garden and managing money. The scales ranging from 0 to 6 for ADL and 0 to 7 for IADL (number of items with reported difficulty) were constructed. To enable us to identify the incident loss of ADLs and IADLs, adults with limitations at wave 4 were excluded from the baseline analysis.

## Statistical analysis

All statistical analysis was performed using Mplus V.8[31] and Stata V.14.[32] We performed incrementally related structural equation models (SEMs): (a) traditional exploratory factor analysis, (b) exploratory bi-factor analysis (EFA), (c) confirmatory factor analysis (CFA), and (d) mediation and moderation analysis.

We first performed a conventional exploratory factor analysis to reveal subfactors of the intrinsic capacity concept using the robust weighted least squares method. Eigen value and scree plot were used to identify number of subfactors to retain. Communalities≥0.3 was selected for minimum loading of an item. We then conducted a bi-factor analysis to examine the possibilities of establishing one general factor (intrinsic capacity). The bi-GEOMIN rotation was implemented that allowed specific subfactors to be correlated with the general factor (intrinsic capacity) and also correlated with each other. The factor structure was further tested in the confirmatory factor analysis. We identified the best fitting model using the inferential goodness-of-fit index in combination

with several descriptive indices: root mean square error of approximation (RMSEA), comparative fit index (CFI), Tucker–Lewis Index (TLI). CFI and TLI values of greater than 0.9 and a RMSEA of less than 0.8 suggest a moderate fit, whereas a CFI and TLI of greater than 0.95 and a RMSEA of less than 0.6 suggest a very good fit.[33] For the bi-factor model, we calculated omega hierarchical coefficients (ωH), because in the bi-factor model the indicators are assumed to be influenced by both the general factor and the specific factors.[34]

We tested the construct validity of the general factor (intrinsic capacity) and specific subfactors in regression analysis. The summary scores for general factor and specific subfactors were generated from CFA by fixing the latent mean to 0 and the latent SD to 1 for each factor. The scores of specific subfactors can be interpreted as the unique contribution of each of the specific domains 'over and above' the general factor (intrinsic capacity). These summary scores were used in the linear regression for testing the construct validity. Simple t-test were performed to examine the statistical difference in the intrinsic capacity score among older persons with or without chronic diseases and results are summarised by age-group and overall population score in two-way boxplot.

Finally, we assessed the predictive validity of the intrinsic capacity score in a mediation model of the direct and indirect relationships of intrinsic capacity and multimorbidity with incident loss of ADLs and IADLs, after controlling for all personal characteristics.[35] PM (ratio of the indirect effect to the total effect) and Rm (ratio of the indirect effect to the direct effect) was calculated to examine the indirect effect size in the mediation analysis.[36 37] For visualising moderation effects, we used the Johnson-Neyman technique.[38] A bias-corrected bootstrap method was used for drawing inference in mediated and moderated analysis.[35]

## RESULTS

### Sample characteristics

Baseline levels of study variables are presented in online supplementary table 1s. Of the 7321 potential participants at baseline, 33% reported either ADL or IADL difficulties and 26% did not provide consent for blood sample analysis (figure 1). A further 28% of the remaining 3581 participants, participants had incomplete information on the independent variables and were also excluded from analysis. The baseline sample therefore comprised 2560 eligible participants. Compared with participants included in the baseline analysis, participants without complete information were older, had a lower education attainment and reported more chronic conditions.

In the follow-up, 91% of baseline eligible participants were re-interviewed. Except education, there was no difference on age, sex, wealth, and multimorbidity status among participants interviewed and not interviewed at the follow-up (online supplementary table s1). No imputation was performed in the analysis and participants with

missing data were excluded, leaving a study sample of 2560 with complete data for the EFA and CFA analysis.

### Bi-factor EFA, CFA and model fit

In the initial exploratory factor analysis, the Kaiser eigenvalues criterion suggested a five-factor model, with 5 factors having Eigen values greater than 1 (ie, 3.1, 2.3, 1.61, 1.23, 1.04). These five factors explained 86% of total variance among the intrinsic capacity indicators. Online supplementary table 2s shows the model fit information for EFA and CFA models tested in the study. One to three factor models provided unacceptable degrees of fit to the data, whereas five factor models provided very good fit, which suggests that intrinsic capacity is a multidimensional construct.

Next, we performed bi-factor EFA under a SEM framework to identify potential modelling problems (eg, sizeable cross loading of intrinsic capacity indicators) and get an early insight on whether primary results of EFA could be replicated with bi-factor model perspectives of multidimensionality. Most items loaded well (≥0.3) on the general factor (intrinsic capacity). Bi-factor EFA revealed one general factor (intrinsic capacity) and five specific subfactors that we labelled cognitive, sensory, vitality, locomotor and psychological (online supplementary table 3s). The model fits the data very well: $\chi^2$=71.2 (df=39), RMSEA=0.012 (90% CI 0.011 to 0.024), CFI=0.99 and TLI=0.99 (online supplementary table 2s). When we examined the factor structure (one factor, second-order, correlated, bi-factor models) in confirmatory factor analysis, the pattern of factor loadings for the bi-factor CFA model showed a clear, simple structure with the five subfactors (figure 2).

Within the bi-factor CFA model, excluding two subfactors (sensory and locomotor), the factor loadings were evenly shared between the general factor and subfactors. However, indicators in the psychological (sleep) and sensory (near vision and distance vision) subfactors had higher loadings on their group factor than on the general factor (intrinsic capacity). This suggests that these two subfactors provide additional information about psychological and sensory capacity, after accounting for the variance of the general factor. The model achieved a good fit for the data: $\chi^2$value=1180.6 (df=89), RMSEA=0.035 (90% CI 0.033 to 0.037), CFI =0.98 and TLI=0.97 (online supplementary table 2s). Indeed, the bi-factor model fit was stronger than for the second order factor model: $\chi^2$=2369 (df=102), RMSEA=0.07, CFI=0.94 and TLI=0.92. Taken together, these findings support this bi-factor model with one general factor representing overall intrinsic capacity and five specific subfactors.

### Reliability of the factor scores

The ωH (hierarchical) coefficient was calculated to understand the reliability of a latent general factor (intrinsic capacity). The ωh value for the general factor was 0.78, and the ωHS (subscore) values for specific factors were. 0.79, 0.80, 0.81, 0.82 and 0.83, respectively. A ωH value

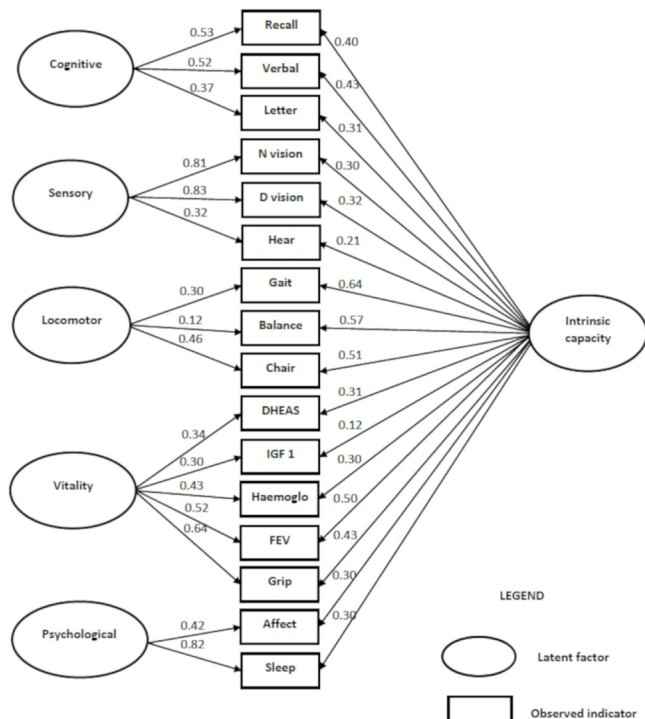

**Figure 2** Bi-factor CFA model of intrinsic capacity. CFA, confirmatory factor analysis; DHEAs, dehydroepiandrosterone; FEV, forced expiratory volume; IGF-1, Insulin-like growth factor 1.

more than 0.7 indicates that the intrinsic capacity total score predominantly reflects a single general factor, suggesting that the total score can be interpreted as a reliable measure of intrinsic capacity. The ωHS more than 0.80 for the subfactor suggests that domain specific scores are equally reliable as the general factor score. Independent of specific factors, the percentage of reliable variance in the score due to the general factor was 72%. This indicates that the intrinsic capacity summary score was a sufficiently reliable measure of the general factor, and added value beyond subfactor scores.

### Construct validity

Factors associated with intrinsic capacity (general factor) and subdomains (subfactors) are presented in the online supplementary table 4s. Lower intrinsic capacity scores were significantly associated with increasing age, female sex, lower levels of education, lower wealth, number of chronic diseases, and number of ADL and IADL limitations. Even after mutual adjustment, all related constructs remained statistically associated with intrinsic capacity (see figure 3). Since all these characteristics have previously been associated with poorer health in older age, these findings support the construct validity of the general factor.

### Associations between intrinsic capacity score and other variables

We used a boxplot of intrinsic capacity score for each chronic condition over three different age groups to display associations between specific chronic conditions

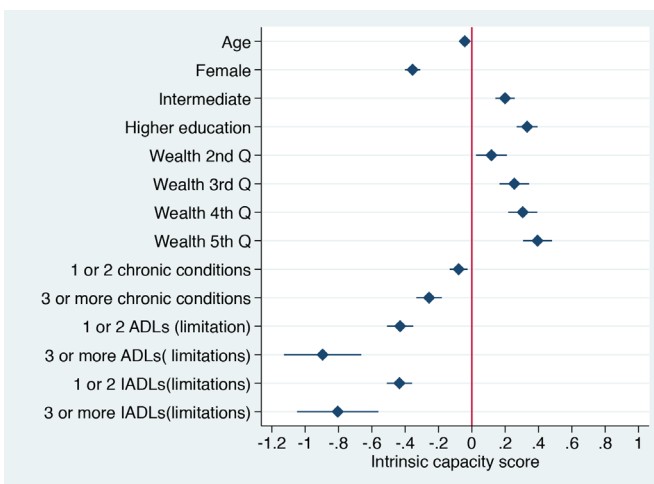

**Figure 3** Construct validity of Intrinsic capacity (mutually adjusted). ADL, activities of daily living; IADL, instrumental activities of daily living.

and intrinsic capacity s (figure 4). Overall, older adults with chronic conditions had statistically significantly lower intrinsic capacity scores (below the mean) than those without chronic conditions and this association was stronger in older age groups. However, the impact of different chronic conditions on the intrinsic capacity scores varied. The greatest impact on intrinsic capacity score was from dementia in the two older age groups. We also examined the intrinsic capacity scores among older people with no chronic conditions in different age-groups. We found that in the absence of any diagnosed chronic conditions, the intrinsic capacity scores tend to decline in higher age-groups. In other words, older people with no diagnosed chronic conditions in higher age-groups (70–79 and 80–100) had significantly lower intrinsic capacity scores than older people in young age-group 60–69 years.

In a separate correlation analysis, we found associations between specific factor scores and various personal characteristics or multimorbidity and these associations were generally consistent with previous research on

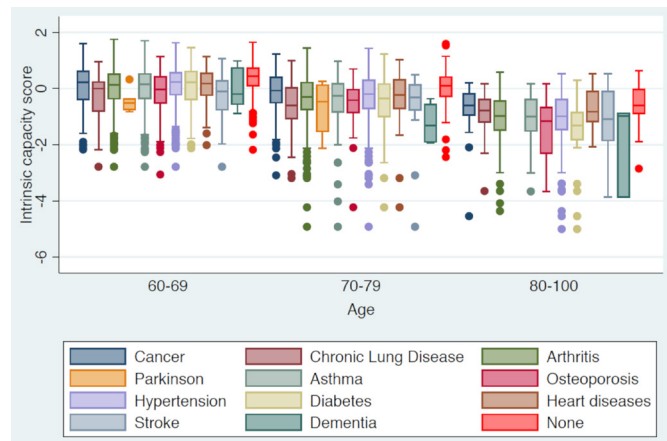

**Figure 4** Intrinsic capacity summary score by chronic health conditions and age-group.

these characteristics (online supplementary table 4s). Cognitive factor scores were negatively associated with increasing age, number of multimorbidities and positively associated with female sex, higher education and wealth (highest quantile). Locomotor scores were negatively associated with age and multimorbidity, and positively associated with higher education, wealth and female sex. Psychological factor scores were negatively associated with increasing age and higher multimorbidity. Higher psychological factor scores were negatively associated with age, female sex and multimorbidity. Vitality subfactor scores were negatively associated with increasing age and multimorbidity, and positively associated with female sex, higher education and higher wealth. The scores of the sensory subfactor were positively associated only with higher education.

### Pathways to care dependence

In the simple mediation model, we tested the direct effect of intrinsic capacity on the incident loss of ADLs and IADLs and the indirect effect through multimorbidity (online supplementary table 5s, supplementary figure 1s).

Intrinsic capacity predicted the incident loss of ADLs and IADLs both directly and indirectly, even after controlling for age, sex, education and wealth. In comparisons of the effect size, the direct effect of intrinsic capacity on IADLs and IADLs was much more prominent than the indirect mediational effect through multimorbidity. In terms of proportion, only a small proportion of the effect of intrinsic capacity on the incidence loss of ADLs (8.7%) and IADLs (5.2%) occurred indirectly through multimorbidity. A bias-corrected bootstrap CI for this direct and indirect effect, which was based on a 10 000-bootstrap sample, was entirely above zero, thus suggesting that these effects are statistically significant.

The results of serial multiple mediators modelling of the relationships between the incident loss of ADLs and

IADLs and personal characteristics, intrinsic capacity scores and multimorbidity are shown in figures 5 and 6.

Both intrinsic capacity score and multimorbidity independently predicted incident loss of ADLs, however only intrinsic capacity independently predicted incident loss of IADLs. Except age, none of the personal characteristics (sex, wealth and education) had a direct effect on incident loss of ADLs and IADLs (online supplementary table 6s). Personal characteristics were strongly associated with both intrinsic capacity and multimorbidity, and the relationship between all personal characteristics (including chronological age) and the incident loss of ADLs and IADLS operated through multimorbidity or intrinsic capacity. A greater proportion of the impact of age on outcomes (30% for ADLs and 39% for IADLs) occurred indirectly through intrinsic capacity than directly (24% for both ADLs and IADLs).

The specific indirect effect of all personal characteristics (age, sex, education and wealth) on the incident loss of ADLs and IADLs through intrinsic capacity was statistically significant (online supplementary table 6s). None of the indirect effect of personal characteristics on incident loss of IADLs operating through multimorbidity was statistically significant. This implies that specific indirect effects of personal characteristics on IADL were mainly transmitted through intrinsic capacity rather than multimorbidity.

In a moderation analysis, after including the interaction term (age*intrinsic capacity), the direct effect of chronological age on incident IADL was not statistically significant (−0.03, p value=0.16). The effect of chronological age on IADL was moderated by a person's level of intrinsic capacity (−0.526, p value=0.004), with the relationship between chronological age and IADL only being significant for people with low intrinsic capacity (online supplementary figure 2s). Similarly, intrinsic capacity moderated the effect of chronological age on incident

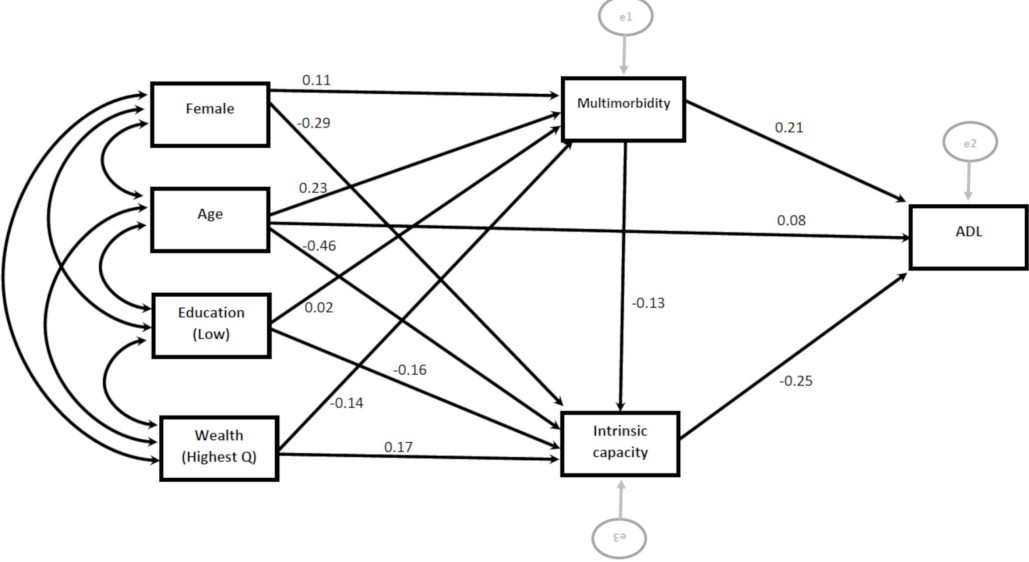

**Figure 5** Direct and indirect effect of characteristics on activities of daily living (ADL).

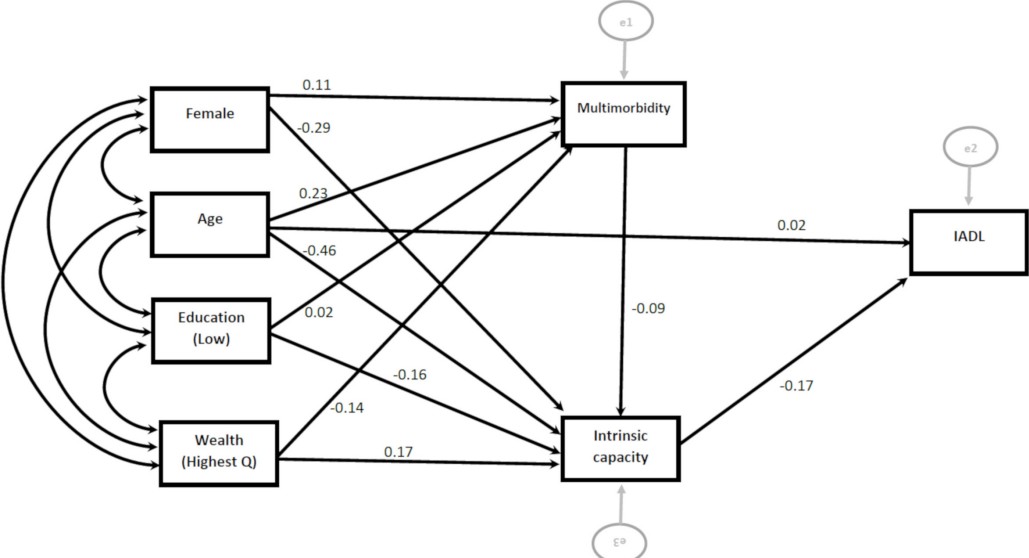

**Figure 6** Direct and indirect effect of characteristics on instrumental activities of daily living.

loss of ADL, after controlling for personal characteristics and multimorbidity (−0.472, p value=0.03).

## DISCUSSION

The WHO model of *Healthy Ageing* provides a transformative framework by which to consider health in older age. Rather than using the entry points of chronological age or disease, the model is built around the concept of intrinsic capacity—all the individual level characteristics that contribute to a person's ability to be and to do what they have reason to value. However, there has been little empirical analysis of the concept and a clear understanding of a possible structure for intrinsic capacity is lacking.

We used a large longitudinal study on ageing to explore the possible structure and predictive validity of the intrinsic capacity concept. We developed a total capacity score for each study participant and found it to be a powerful predictor of incident care dependence, even after accounting for chronological age and the presence, or number, of key health conditions. Factor analysis suggested a structure comprising five subfactors—psychological, sensory, cognitive, vitality and locomotor. This may provide a frame for the construct that is readily applicable to research and clinical practice.

These findings suggest that the intrinsic capacity concept has an empirical rigour and captures information beyond that generally considered in research or clinical practice. It also suggests that multiple domains of capacity can be aggregated into a meaningful overall measure of health status. If confirmed by future studies, these findings have a number of significant implications. For example, routine monitoring of intrinsic capacity might enable clinicians to flag when trajectories of capacity in the second half of life are veering off normal—a similar approach to the way child development charts currently guide paediatric practice.[39] A recent meeting of expert

geriatricians convened by WHO confirmed that this would be useful, particularly if score changes could be interpreted in ways that have clinical relevance.[40] The factor structure of capacity identified in this analysis may provide a framework that achieves this by allowing clinicians to identify and address the drivers of any changes.

Measurable trajectories of capacity may also be useful as research outcomes of interest. As continuous measures that can be monitored at multiple time points, they allow a more nuanced and powerful analysis than approaches that use crude categorical measures of late life events such as mortality or incident loss of ADLs and IADLs.[41] Moreover, if information was available on trajectories of capacity across the full second half of life, this may facilitate the identification of mid-life influences on late life health which may be amenable to intervention. This is likely to become more feasible with the rapid development of wearable and communications devices which are already generating large amounts of relevant and routinely collected data. Appropriate algorithms could be developed to process this information to describe trajectories of capacity that could inform self-management, clinical practice and research.

Using trajectories of capacity as a research outcome may also allow better comparison between the impacts of interventions for different conditions. Furthermore, as medicine becomes increasingly personalised and precise, better information is needed on how different subpopulations may respond to specific interventions.[42] Stratifying by intrinsic capacity may provide a useful way of identifying the groups for which interventions are most effective and may be more appropriate than categorisation by chronological age or comorbidity.

One critical issue requiring further work is that not all five subfactors appear to operate at the same level. The cognitive, locomotor, sensory and psychological sub factors can be thought of as overt expressions of capacity.

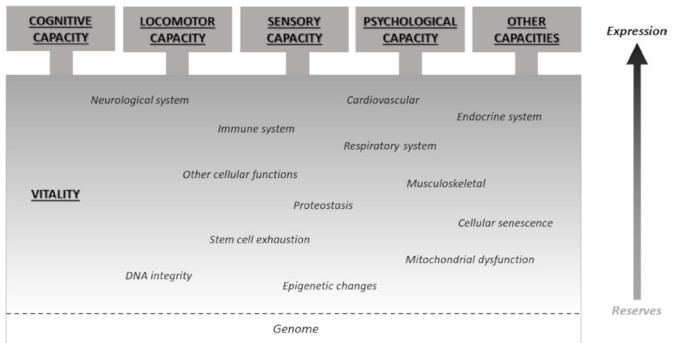

**Figure 7** Conceptual frame for the construct of intrinsic capacity.

On the other hand, DHEA, IGF-1, haemoglobin and FEV (included in the vitality subfactor) are elements of the biologic systems that underlie these overt manifestations of capacity.[6] Grip strength, the other characteristic loading to the vitality subfactor, can also be considered a marker of broader underlying factors such as nutritional, immune and hormonal status, and in this sense it is interesting that it loaded separately to locomotor capacity.[43 44]

The vitality subfactor interacts strongly with the other subfactors and part of the contribution it makes to the intrinsic capacity score is through the influence it has on these overt expressions of capacity. However it also loaded independently to the general factor (intrinsic capacity).

One possible conceptual frame for these relationships starts with a vitality domain describing variance in the complex and dynamic biologic systems which sustain life and functioning. When accumulated deficits in these systems reach a certain point they become manifest in the overt losses of capacity that are commonly associated with ageing. However, deficits in these systems that may not yet be expressed in overt manifestations are also likely have implications for the ability of the individual to retain their level of functioning. This residual is consistent with the notion of physiologic reserves or physiologic 'resilience'. A total measure of vitality may thus capture an individual's 'biological age'.

Figure 7 shows how these domains might hypothetically relate. We have included a space for specific expressed capacities not captured in the four domains identified in our analysis (for example continence and speech). Within the vitality construct we have included cellular level characteristics as well as the contribution of higher physiologic systems. This is consistent with our analysis but also suggests at how characteristics not assessed (see strengths and weaknesses) might be considered in a conceptual frame for intrinsic capacity.

A second issue is that all the more overt capacities also interact. This could be explained using the conceptual frame proposed above since the biologic drivers of these capacities are shared. This finding is also consistent with research and clinical experience which suggest that decrements in one domain of capacity may have clinically relevant impacts on other domains. For example, gait speed can be influenced by simultaneously drawing on an individual's cognitive capacity (eg, by being asked to count backwards). These complex interactions may indeed provide the opportunity for 'stress' testing of scores in any single domain.[45]

However, the combined score we have calculated takes no account of thresholds that may exist within each subfactor. For example, cognitive capacity may fall to the point where it becomes impossible for an individual to survive without appropriate care and support, even though they may retain perfect capacity in each other domain and thus retain a relatively high total capacity score. This emphasises the need to assess the multiple dimensions of capacity to fully assess the clinical importance of changes in total score.

## Strengths and limitations of study

A strength of this study is that it is a large, nationally representative sample of older people living in England with good follow-up. Unlike approaches that use a composite total score which assumes that each indicator or measure contributes equally to the general factor (ie, intrinsic capacity), we used the bi-factor model scores that represents a pure measure of the underlying latent trait of interest, after controlling for all five specific subfactors.[46] Hence, a theoretically error-free score was used in all analysis to study the unique contribution of intrinsic capacity and its components in the prevention of care dependence.

Second, the longitudinal nature of the study allowed us to examine the direction of causality. Third, most of the indicators of intrinsic capacity were measured using objective performance tests, limiting opportunities for response or interviewer bias.

However, it is important to note that the measures included in the ELSA study are neither complete nor random. They were chosen to inform specific research questions of interest to the investigators, rather than to create an overall measure of intrinsic capacity.[12] Additional variables may alter the total capacity scoring and the factor structure. Nevertheless, since these questions largely draw on existing knowledge and research priorities, they cover aspects of most domains that might be conceptualised within the notion of capacity. Some potential components of capacity cannot be readily measured objectively (for example energy levels). Others require complex assessments that are beyond the scope of primary care or population-based research (for example, continence, cardiovascular capacity). Changes in other important attributes like the capacity for speech are important but less common. A number of key biomarkers, for example telomere length and immune function, were also missing from this dataset. Thus, while the set of indicators considered in this analysis can be considered relatively comprehensive, they are not complete in their ability to measure all aspects of capacity. Moreover, while we attempted to limit analysis to objective measures, the only data available on sensory and psychological capacities was through self report. This should not have had a

significant impact on the construct of capacity, but may have had a marginal influence on the longitudinal analysis we undertook.

Despite carefully accounting for potential confounders, measurement error in their assessment, particularly the difference between participants who could and could not provide complete information on all exposure measures, may have biased associations. Also, the number of chronic diseases included in the analysis are limited, hence there is possibility of residual confounding.

Our findings are, however, consistent with previous research on the subfactors that were included in our analysis. Several longitudinal studies have shown strong predictive validity of cognitive (namely memory and executive function),[47 48] locomotor (gait or chair rise),[49–51] sensory (vision and hearing),[23 52–54] vitality (hand grip strength or FEV),[55–58] and psychological[59] indicators in relation to incident loss of ADL and IADL. Studies have also demonstrated associations between indicators of intrinsic capacity and survival. In particular, studies of locomotor and cognitive functions have shown that these indicators are predictors of premature mortality in community dwelling populations.[60–62] Yet, traditionally, these characteristics have often been considered independently. The intrinsic capacity concept provides a vehicle for assessing how they relate to each other and a possible approach to better quantify ambiguous notions such as 'health' in older age into research and clinical practice.[40 63]

## CONCLUSIONS
Measurement of intrinsic capacity is feasible with commonly used measures and appears to provide useful predictive information on an individual's subsequent functioning. The proposed general factor and subfactors structure may contribute to a transformative paradigm for future research and clinical practice.

**Acknowledgements** The views expressed in this paper are those of the authors and do not necessarily reflect the views of WHO. The authors would like to thank the ELSA participants, the ELSA researchers and the UK Data Service for enabling the use of ELSA data for this analysis.

**Contributors** JRB conceived of the research, oversaw analysis and was responsible for final drafting of the paper. ATJ undertook all analyses, reviews of related literature, and contributed to drafting of the paper. MC contributed to conceptualisation, reviews of related literature and drafting of the paper. IAC contributed to conceptualisation, reviews of related literature and drafting of the paper. All authors reviewed and approved the final manuscript submitted for publication.

**Funding** The authors have not declared a specific grant for this research from any funding agency in the public, commercial or not-for-profit sectors.

**Competing interests** None declared.

**Patient consent for publication** Not required.

**Ethics approval** Ethical approval for ELSA was obtained from NHS Research Ethics Committees under the National Research and Ethics Service (NRES), and participants gave full informed written consent for participation. More information on ELSA can be found at http://www.ifs.org.uk/elsa/documentation.php.

**Provenance and peer review** Not commissioned; externally peer reviewed.

**Data sharing statement** No additional data are available. However, ELSA dataset and information on all currently archived can be freely accessed through the UK Data Archive (https://www.elsa-project.ac.uk/availableData).

**ORCID iDs**
John R Beard http://orcid.org/0000-0002-8557-0242
Islene Araujo de Carvalho https://orcid.org/0000-0003-3842-9999

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
