## [Reviewer comments · BMJ Open]

This paper was submitted to a another journal from BMJ but declined for publication following peer review. The authors addressed the reviewers' comments and submitted the revised paper to BMJ Open. The paper was subsequently accepted for publication at BMJ Open.

ARTICLE DETAILS

TITLE (PROVISIONAL)	THE STRUCTURE AND PREDICTIVE VALUE OF INTRINSIC CAPACITY IN A LONGITUDINAL STUDY OF AGEING
AUTHORS	Beard, John; Jotheeswaran, AT; Cesari, Matteo; Araujo de Carvalho, Islene

VERSION 1 – REVIEW

REVIEWER	REGINSTER Jean-Yves President and Chair Department of Public Health, Epidemiology and Health Economics University of Liege – Liege, Belgium
REVIEW RETURNED	16-May-2018

GENERAL COMMENTS	This paper deals with a very important component of the efforts made to improve healthy aging. The identification of the major dimensions of intrinsic capacity as a research agenda, is on the table since several years and this paper provides a major contribution to its better understanding and implementation. The results presented here come from a major international trial which is extensively and appropriately analyzed. The conclusions are sound and fully supported by the results. The only point is that the authors should comment on the reason why no quality of life assessments were made during this study or if they were made, why they were not analyzed. Most likely, self-perception of health status or self-perception of quality of life by the patients play a significant role in their amount of intrinsic capacity and on their resilience to external stressors. This might be a limitation of the study which should be better developed and at least acknowledged. Apart of this, I think that this contribution is a major step towards better understanding how to cope with ageing patients and to the maintenance of the healthy aging dimension.
--

REVIEWER	Rachel Cooper Senior Lecturer UCL
-----------------	---

GENERAL COMMENTS

This research article uses data from the English Longitudinal Study of Ageing to demonstrate a method by which intrinsic capacity, a concept which is central to the WHO's World report on ageing and health, could be operationalised. The concept of intrinsic capacity is a useful one and does have the potential, as part of the WHO's framework for healthy ageing, to help drive forward research on ageing. However, as the authors highlight, a challenge has been in identifying how best to operationalise this concept for application in research and clinical settings. While attempts to address this are therefore welcomed, this article would benefit from being placed in the context of other work of relevance; the WHO have done great work in this area but so too have others (more details below). It is also not clear how relevant this will be to a general readership given some of the issues highlighted below.

Some of the standard requirements of a BMJ submission are not included which is surprising; in my own experience these are requested before the editors are willing to consider the manuscript for review. For example, the submission does not appear to include a structured abstract, a public and patient involvement statement or summary boxes on what is already known on this topic and what this study adds (the 'Research in Context' box included is a requirement for the Lancet).

While the overall design of the study appears to be adequate to address the research question outlined, I am not qualified to comment on the details of the statistical analyses. For this reason, I think the article would benefit from a statistical review.

Specific comments

1) In the introduction the authors state that 'The most commonly used indicators of functioning in older age – IADLs or ADLs – can sometimes fail to distinguish between capacity and ability.' ... 'These measures are also generally only sensitive to very significant losses of functioning.' Similar statements are also made in the discussion. I was surprised by this as it has long been recognised that self-reported measures of functioning (such as reports of difficulties with IADLs and ADLs) have a number of important limitations. It was for this reason that researchers including Jack Guralnik (previously of the NIA) began devising and promoting the use of performance based measures of function which overcome many of these limitations and facilitate research on ageing as early as the late 1980s (see Guralnik et al JGMS 1989;44:M141-6 and Guralnik and Ferrucci Am J Prev Med 2003;25:112-21). As a result of these previous research efforts performance based measures of functioning are now widely used and it could be argued are more often a focus of research, especially in studies taking a life course approach to the study of ageing, than IADLs and ADLs. I draw here on examples on physical functioning but I believe the same is true for cognitive functioning.

2) I am pleased that the WHO model recognises the importance of a life course approach to the study of ageing (second paragraph of introduction). However, it would be helpful, in allowing readers to place this work in context, to see some acknowledgement of the research that I assume the WHO have drawn on or at least been aware of in developing this model; for example, for many years

Diana Kuh, Yoav Ben-Shlomo and colleagues (including myself) have developed and promoted the application of a life course approach to health and ageing with the study of trajectories of function across life central to this (see for example, Ben-Shlomo et al Int J Epidemiol 2016;45:973-988, Ferrucci et al JGMS 2016;71:1184-94 and Kuh et al (editors) 'A life course approach to healthy ageing' (2014, OUP)). Other researchers have also been developing and promoting these ideas (see for example Moffitt et al JGMS 2017;72:210-5; <http://athlosproject.eu/>; <https://www.lifepathproject.eu/>)

3) In relation to points 2 and 3, it is important to acknowledge the wide range of measures that have already been developed to study functional trajectories across life – the short physical performance battery is only one. See for example, the work done to develop a range of functional tests that can be assessed across life (from age 3 to 85+) as part of the NIH toolbox initiative (nihtoolbox.org and Neurology 2013;80 (11 Supplement 3)).

4) While the authors make a strong case for developing 'a clearer conceptualisation of the intrinsic capacity construct' it is not entirely clear why the authors have selected the specific measures they have for inclusion in their score. To what extent was the selection of measures based on the pre-existing conceptual framework and to what extent was it based on data availability and pragmatic decisions that needed to be taken related to this? Was any consideration given in selecting these measures to how best to ensure that it would be possible to replicate a similar construct in other datasets and study populations?

5) It would be helpful if the authors could provide clearer justification for deciding to relate their total intrinsic capacity score to subsequent IADL and ADL loss as a test of its validity. Were other important outcomes, such as survival and institutionalisation, also considered? Please also make it clear why multi-morbidity and 'personal characteristics' were included in analyses.

6) Please confirm why waves 4 and 5 of ELSA were selected for use in analyses when 7 waves are available.

7) Did the authors consider using data from another wave at which all relevant variables were available to test the derivation of their total intrinsic capacity score and check that similar results were found at other time points?

8) In the introduction, the authors highlight that it can be difficult to distinguish intrinsic capacity from environmental adaptation. However, some of the measures included in the total intrinsic capacity score may capture environmental adaptations. For example, questions on hearing status take account of hearing aid use and questions on sleep disturbance do not take account of environmental adaptations that may have been made to account for underlying sleep difficulties.

9) It is reported on page 5, that 6238 participants aged 60+ were included in ELSA wave 4. However, only 2532 were included in the analyses presented. This large drop in N is of concern not only because it is likely to have introduced bias but also because it highlights practical problems in deriving the intrinsic capacity score. Please could the authors clarify why such a large drop in N

occurred and comment on similarities and differences in the characteristics of the people they were able to include in their analytical sample and those they had to exclude and the implications of this. The authors should consider methods that may allow them to minimise loss of participants from the derivation of the score.

10) As it is already well established that all of the variables included in the intrinsic capacity score are related to subsequent risk of morbidity, institutionalisation and premature mortality it is perhaps not surprising that the total capacity score was found to be a powerful predictor of incident care dependence. I think the authors therefore need to make a much clearer case for what their study really adds.

11) To be confident that this construct is readily applicable to research and clinical practice (as suggested on page 8), evidence would be required of how it performs in other general community-dwelling and clinical populations. In addition, before it could be applied to 'assist research into the early determinants of functional change' it would be necessary to understand how the score performs in younger populations given only people aged 60+ were included in these analyses. Do the authors have plans for this?

12) Time for assessment in both clinical and research settings is limited. A number of different measures are included in the total intrinsic capacity score and many of these are highly correlated. How confident are the authors that each of these measures provides added value? Before routine assessment of such a measure and all its many components could be considered, evidence would be needed that each component was required.

13) Related to point 12, capturing data on biomarkers tends to be more invasive, time consuming and expensive. In addition, there is limited evidence that measures such as telomere length do add predictive value for future outcomes if included in models with measures that capture function at the individual level (see for example, von Zglinicki BMJ 2012;344:e1727 and Martin-Ruiz et al Mech Ageing Dev 2011;132:496-502). It would therefore be helpful to see the authors comment on whether the inclusion of blood-based biomarkers is likely to add value and be necessary especially in light of its associated costs.

14) A number of other composite scores have been developed that aim to capture different aspects of health and function in older people and which are proposed as tools for monitoring patients and, screening populations to identify those who are at high risk of poor outcomes in later life. For example the frailty index, which is now being applied in clinical settings (see for example, Clegg et al Age Ageing 2016;45:353-60). While these have drawn on different conceptual frameworks, to what extent do they really differ from the intrinsic capacity score especially in terms of their clinical utility?

15) Page 9, it is not clear how the longitudinal nature of the study allowed the authors to examine the direction of causality. In addition, the fact that some of the measures used were self-reported (e.g. for sensory function and sleep) does need to be acknowledged and the implications considered.

	16) Please consider citing systematic reviews and meta-analyses on the associations between functional measures and survival in place of references 50-52 (e.g. Studenski et al JAMA 2011;305:50-58; Cooper et al BMJ 2010;341:c4467; Calvin et al Int J Epidemiol 2011;40:626-44) 17) Caution is required in promoting the use of a 'holistic outcome measure' to 'assist research into the early determinants of functional change' especially as evidence suggests that different functional measures have different underlying aetiologies whereby important lifetime risk factors specific to one component may not be identified if only the total score is considered in analyses. Minor points 1) On page 2, when describing the chair-stand test, the authors refer to the eligibility criteria for the balance test. Please correct this. 2) On page 3, it would be more accurate to report that a trained nurse collected blood samples which were then analysed to generate biomarker data (rather than reporting that trained nurses collected biomarker data). 3) In listing the measures used in analyses please make it clear which are considered for inclusion in the intrinsic capacity score and which are factors that have been used in subsequent analyses. 4) Wherever possible please include results tables in the main paper rather than as supplementary data. For example, if BMJ article requirements allow it, it would be helpful for supplementary table s1 to be included as table 1 of the main paper. 5) On page 7, when referring to the 'direct effect' it is important to make it clear that this is the effect that is not explained by multimorbidity i.e. it is possible that this is mediated by other factors not considered. 6) Please consider adding footnotes to the figures to help readers understand what is shown without having to refer to the text. 7) In table 1s it would be helpful to see the distributions of the variables that are included in intrinsic capacity score and for this information to be stratified by sex (given there are sex differences in the distributions of many of these variables).
--	---

REVIEWER	Professor Cyrus Cooper Director MRC Lifecourse Epidemiology Unit
REVIEW RETURNED	05-Jun-2018

GENERAL COMMENTS	This interesting manuscript takes forward the concept of intrinsic capacity, central to the World Health Organisation report on ageing and health. In this novel conceptual model, focus is shifted from the absence of disease, to a capabilities based approach which aims to maintain the functional ability for older people. This functional ability, is determined by the intrinsic capacity of the individual, the environments in which they live, and the interaction between the individual and their environment. Intrinsic capacity is
--

	defined as the combined grouping of physical and mental capacities that an individual can draw on at any point in the lifecourse, but the individual components of intrinsic capacity and their measurement, remained subjects for further study. In this manuscript, the authors use the English longitudinal study of ageing to evaluate the capacity of a variety of measures to be integrated within five broad domains of intrinsic capacity (sensory, cognitive, psychological, locomotor and vitality) as well as their capacity to predict impairment in activities of daily living (ADL and IADL). The methods used are rigorous, and I particularly liked the approach in which factor analysis was tiered in theories, using traditional exploratory factor analysis; exploratory bi-factor analysis; a confirmatory subsequent analysis; and then mediation/moderation analysis. The data provide strong support for these five domains of intrinsic capacity, and a degree of face, construct and content validity. The conclusion that measurement of intrinsic capacity is feasible with a series of commonly used measures, and appears to provide useful predictive information on an individual subsequent functioning, is strongly supported by the findings. I have little doubt that the conclusions and their overarching impact on clinical medicine for the elderly, will be substantial. On this basis, the findings are well worth publishing in the BMJ. The only fault that I had, was whether the manuscript was so detailed in the description of the statistical methodology; the exposition of the results; and the subsequent pathway analysis to care dependence; that some of the more technical aspects might be better incorporated within a web based annex, with the principal findings streamlined for publication in the journal itself. Finally, knowing the competition for such papers in the BMJ, if the manuscript does not meet the publication threshold, I wonder whether an editorial, or an abbreviated review of the topic, would be appropriate for the readership. I have little doubt that this conceptual framework will dominate the future of comprehensive geriatric assessment among healthcare systems worldwide, and has the great advantages of doing so regardless of underlying population wealth.
--	---

REVIEWER	Sandhi Maria Barreto Professor Universidade Federal de Minas Gerais, Brazil
REVIEW RETURNED	13-Jun-2018

GENERAL COMMENTS	The article is very well designed, the study subject is of great social and clinical relevance, and the methods are appropriate to data structure and study objectives. I congratulate the authors. I have only a few questions to make to the authors. Care dependence was assessed using self-reported limitations in the Basic Activities of Daily Living (ADL) and Instrumental Activities of Daily Living (IADL). The description suggests that the authors used scores based on no/yes (0/1) answer only, with no consideration for level of difficulty. Why not? The multimorbidity score was based on information on doctor diagnosis of 14 conditions, some of which are more interrelated than others, as the cardiovascular ones. On the other hand, they just include cancer as a broad category. Was the choice of the 14 conditions based on what was available in the ELSA cohort or it was based on a pre-selection of conditions that matter. What the
--

	authors recommend as a good repertoire of medical conditions to add to this score. The list and number of conditions included will influence categorization, especially because the multimorbidity variable is grouped as none, 1 or 2, and 3 or more conditions. The authors did not mention, but I presume that the presence of physical defects, including upper or lower limb loss or impairment, poor manual dexterity, and damage to one or multiple organs of the body should be considered either separately or as part of the multimorbidity score. In the analysis, the authors state that “No imputation was performed in the analysis and participants with missing data were excluded, leaving a study sample of 2532 with complete data for the EFA and CFA analysis” . Considering the initial sample 6238 individuals at Wave 4, it means that only 40,6% of the sample was included. Thus, it cannot be discarded some differential losses related to physical and mental capacity items that may impact the overall analysis. Table 1 should present the distribution of characteristics of the individuals that contributed to the analysis (n=2532) as well and p-value to indicate differences with the total sample (n=6238). Finally, I think the authors make a very good point when they say “...the combined score takes no account of thresholds that may exist within each subfactor. For example, cognitive capacity may fall to the point where it becomes impossible for an individual to survive without appropriate care and support, even though they may retain perfect capacity in each other domain and thus retain a relatively high total capacity score.” I think the authors could develop this a bit further and perhaps indicate that the use of a multidimensional approach would not be adequate (or necessary) when a clearly uncapable health condition (physically or mentally) exists.
--	--

REVIEWER	Martin Connolly Professor of Geriatric Medicine University of Auckland
REVIEW RETURNED	05-Jul-2018

GENERAL COMMENTS	This is an interesting paper in an area of great and increasing importance. The paper is well written but this reviewer has some concerns about the methodology particularly regarding the collection of the baseline data. These are concerns detailed below. This reviewer recommends an independent statistical review of the paper as the statistical methodology is complex and rather specialised. Major Concerns  1. Paragraph 2; line 4: All subjects ‘signed fully informed consent’. Though this is clearly appropriate it does censor the study data particularly as one of the major covariates in the comorbidity analysis was dementia. The authors should comment on this in the discussion section as an important weakness of the study (particularly given that dementia was such a predictive covariate).
--

2. Page 3; lines 5-9: This reviewer is concerned about the apparent lack of understanding of lung function measures. It appears that the measure that the authors have conducted is forced expiratory volume in one second (my underlining) and not forced expiratory volume (line 5). Further to this they (line 8) state that they have conducted forced expiratory volume in one minute which is clearly incorrect. Equally importantly they (line 8) state that they have used the 'highest technically satisfactory measure' of forced expiratory volume. They do not state which criteria or guideline they base this latter assertion upon. Forced expiratory volume in one second has very clear international acceptable guidelines for its assessment. It is unclear to this reviewer whether the authors are aware of these and whether they following them.
3. Page 3; lines 20-27: I am very concerned by this paragraph. Hearing impairments in particular are usually under-reported by older people (and probably by younger people as well). What validation was there for such self-reported measures of hearing? I am less concerned about the self-reported measures of vision.
However, once again the authors are working from data that is available to them and can do no more - but (also once again) they should cite this as limitation of the study in the discussion section.
4. Page 3; lines 32-35: The authors need to provide a reference for their assertions regarding taking animal naming as a measure of executive functioning and taking letter cancellation as a measure of processing speed.
5. Page 4; line 7: Why were myocardial infarction and angina separate measures of multimorbidity? They represent the same disease. Later on in the paper (eg Figure 3) these and another comorbidity in the list quoted at the top of page 2 (congestive heart failure) seem to be lumped together as 'heart disease'. The authors need to be clear on whether each of these covariates (myocardial infarction, congestive heart failure and angina) have been analysed separately or individually. If the latter they need to justify the separate analyses for myocardial infarction and angina.
6. Page 5: The authors do not define the abbreviations EFA and CFA at any point.
7. Page 7; first full paragraph: This is the beginning 'so what' section of the paper, ie what is the usage of intrinsic capacity? As such I do not believe that Table 5s and Figure 1s should be supplementary to the paper. They should be part of the main published article.
8. The Discussion Section is particularly powerful (most especially the third, fourth and fifth paragraphs on Page 8). However, (page 9; lines 6-8) whilst technically accurate needs to take account of my comments regarding the potential flaws in the methodology as mentioned above.
9. Figures 1, 4 and 5: A key is needed explaining what the numbers adjacent to the lines within the figures mean.

Minor Points

1. Page 1; line 22: I would suggest that the words 'can sometimes' are deleted.
2. Page 2; line 55: I assume that when the authors refer to 'average' are they in fact referring to 'mean'? If so, they should say so.
3. Page 3; line 17: Note the inaccurate spelling of the word haemoglobin.

	4. Page 6; paragraph headed 'construct validity': I would suggest that this paragraph is moved above the paragraph headed 'reliability of the factor scores' 5. Page 6; line 32: By 'these conditions' this reviewer presumes the authors mean the comorbidities. This should be stated specifically as it is not stated in the bolded paragraph heading.
--	---

VERSION 1 – AUTHOR RESPONSE

RESPONSES TO REVIEWER COMMENTS

Reviewer: 1

“reason why no quality of life assessments were made during this study or if they were made, why they were not analyzed. Most likely, self-perception of health status or self-perception of quality of life by the patients play a significant role in their amount of intrinsic capacity and on their resilience to external stressors. This might be a limitation of the study which should be better developed and at least acknowledged”.

Response

In the WHO framework, “health status” is consistent with the concept of functional ability. This comprises one’s intrinsic capacity, their environment and the interaction between these characteristics. This analysis is limited to the intrinsic capacity construct. Later analysis could naturally extend to self perceptions of health status, i.e. of functional ability, but this is not the focus of our analysis. Moreover, we attempted, where possible, to limit our analysis to directly observable measures. If the construct we are proposing is confirmed, there are numerous psychological assessments, that could be incorporated into the domain of psychological capacity.

Reviewer: 2

While attempts to address this are therefore welcomed, this article would benefit from being placed in the context of other work of relevance; the WHO have done great work in this area but so too have others (more details below). It is also not clear how relevant this will be to a general readership given some of the issues highlighted below.

Specific comments

1) In the introduction the authors state that ‘The most commonly used indicators of functioning in older age – IADLs or ADLs – can sometimes fail to distinguish between capacity and ability.’....’These measures are also generally only sensitive to very significant losses of functioning.’ Similar statements are also made in the discussion. I was surprised by this as it has long been recognised that self-reported measures of functioning (such as reports of difficulties with IADLs and ADLs) have a number of important limitations. It was for this reason that researchers including Jack Guralnik (previously of the NIA) began devising and promoting the use of performance based measures of function which overcome many of these limitations and facilitate research on ageing as early as the late 1980s (see Guralnik et al JGMS 1989;44:M141-6 and Guralnik and Ferrucci Am J Prev Med 2003;25:112-21). As a result of these previous research efforts performance based measures of functioning are now widely used and it could be argued are more often a focus of research, especially in studies taking a life course approach to the study of ageing, than IADLs and ADLs. I draw here on examples on physical functioning but I believe the same is true for cognitive functioning.

Response: We apologise for the confusion this introduction has created and acknowledge the extensive previous work that the reviewer has highlighted. We have reframed this section to clarify our intentions and also underline what our analysis adds to the research base identified by the reviewer. We have also included the various references the reviewer suggests as well as others to acknowledge this work.

The point we were trying to make, but obviously did not express clearly in these abbreviated comments, was that there are two common problems in the application of assessments of overall functioning: failure to distinguish between the role of the individual and the role of the environment, and difficulty in assessing the overall capacity of the individual and in understanding how this overall capacity is composed are structured.

The WHO Healthy Ageing model attempts to do this by describing the total individual level attributes that contribute to a person's ability to function as their "intrinsic capacity", while proposing that their "functional ability" (their ability to be and to do the things they value) is determined by this capacity plus the influence of the environment they inhabit plus the interaction between the individual and their environment. In developing this model, we undertook a comprehensive review of existing literature, including the work identified by the reviewer and also consulted top experts in the field. The details of this effort are described in the World report and also in the research in context box. More information on the limitations of current approaches is included in chapter 2 and 3, World Report on Ageing and Health <http://www.who.int/ageing/events/world-report-2015-launch/en/> .

Of course, many rigorous performance based measures of functioning are now used, but generally these disaggregate intrinsic capacity into specific capacities (for example cognitive or physical as the reviewer suggests). Much less work has been done considering the relationship between these capacities or how they contribute to an individual's overall intrinsic capacity. For example, the Guralnik work identified by the reviewer uses a Nagi approach in which it identifies "functional limitations", which are somewhat analogous to our concept of intrinsic capacity. However, the way these researchers construct this entity is somewhat ad hoc. Our work is the first we are aware of to try to empirically describe this emergent property and to distinguish the functional subdomains from disease related characteristics.

Moreover, our interest here is not to suggest new ways of measuring these variables – in fact we are relying on existing performance based measures to develop a conceptual framework that describes the structure of these total individual level attributes, how they relate to each other and whether the total emergent property that is intrinsic capacity has significant prognostic value.

This conceptual framing is particularly important from a life course perspective since if, for example, intrinsic capacity is structured as our analysis suggests, one would expect this structure to be consistent across the life course, although the measures you may use to assess these capacities would need to change. For example while cognitive tests used in child development are not the same as those used to test cognitive decline in older age, the domain is likely to be consistent and an individual's position against the normal range of cognitive capacity at any age can be assessed. This is similar to the methods used by the NIH Toolbox, although the latter lacks the conceptual framing we are trying to establish (but it is consistent with it)

2) I am pleased that the WHO model recognises the importance of a life course approach to the study of ageing (second paragraph of introduction). However, it would be helpful, in allowing readers to place this work in context, to see some acknowledgement of the research that I assume the WHO have drawn on or at least been aware of in developing this model; for example, for many years Diana Kuh, Yoav Ben-Shlomo and colleagues (including myself) have developed and promoted the

application of a life course approach to health and ageing with the study of trajectories of function across life central to this (see for example, Ben-Shlomo et al *Int J Epidemiol* 2016;45:973-988, Ferrucci et al *JGMS* 2016;71:1184-94 and Kuh et al (editors) 'A life course approach to healthy ageing' (2014, OUP)). Other researchers have also been developing and promoting these ideas (see for example Moffitt et al *JGMS* 2017;72:210-5; <http://athlosproject.eu/>; <https://www.lifepathproject.eu/>)

Response:

Indeed, we have engaged closely with Profs Ben-Shlomo and Kuh, as well as a number of participants in the Athlos Project, on aspects of our work. We have added comments to highlight this substantial body of work as well as a number of references including those the reviewer suggests.

3) In relation to points 2 and 3, it is important to acknowledge the wide range of measures that have already been developed to study functional trajectories across life – the short physical performance battery is only one. See for example, the work done to develop a range of functional tests that can be assessed across life (from age 3 to 85+) as part of the NIH toolbox initiative (nihtoolbox.org and *Neurology* 2013;80 (11 Supplement 3)).

Response: As stated above. Please note that the objective of this paper is not to validate or recommend a specific test for IC. The objective is to test theoretical concept and public health utility of the concept. The NIH Toolbox is a wonderful resource that was created with the aim "to develop a set of state-of-the-art measurement tools" of "neurological and behavioural function". The domain structure (which is not inconsistent with our own) was established after literature review and consensus of participants. Unlike our work, the structure was never designed to provide a summative measure of capacity, was not based on empirical assessment, and did not explore the relationship between domains.

4) While the authors make a strong case for developing 'a clearer conceptualization of the intrinsic capacity construct' it is not entirely clear why the authors have selected the specific measures they have for inclusion in their score. To what extent was the selection of measures based on the pre-existing conceptual framework and to what extent was it based on data availability and pragmatic decisions that needed to be taken related to this? Was any consideration given in selecting these measures to how best to ensure that it would be possible to replicate a similar construct in other datasets and study populations?

Response:

This is a good point which we attempted to address in the discussion. We have extended this discussion in the limitations section.

5) It would be helpful if the authors could provide clearer justification for deciding to relate their total intrinsic capacity score to subsequent IADL and ADL loss as a test of its validity. Were other important outcomes, such as survival and institutionalisation, also considered? Please also make it clear why multi-morbidity and 'personal characteristics' were included in analyses.

Response: We wanted to use an outcome measure of functioning independent of the measures included in the construct. We did not test survival as we anticipated the study would lack the power over the relatively short study period to demonstrate a relationship. Institutionalisation is not just determined by an individual's capacity but also by the availability of an institution which made it problematic as an outcome variable, making it inappropriate for this analysis. We have included more discussion on this reasoning in the discussion section.

6) Please confirm why waves 4 and 5 of ELSA were selected for use in analyses when 7 waves are available.

Response: To cut down the cost, ELSA applied planned missing-data design, in which some parts of information about respondent or certain biomarkers are purposely not collected. https://www.elsa-project.ac.uk/uploads/elsa/docs_w7/ELSA%20content%20waves%201%20to%208%20v1.0.pdf . The wave 4 was selected due to availability of all relevant biomarkers of intrinsic capacity. Wave 5 was chosen, to test short term predictive validity and to reduce number of missing data at the follow-up. We have included this reasoning in the methods section text.

7) Did the authors consider using data from another wave at which all relevant variables were available to test the derivation of their total intrinsic capacity score and check that similar results were found at other time points?

Response: As state above, not all biomarkers are collected in all waves. For example, Word-finding (verbal fluency) and Letter cancellation (accuracy and speed of mental processing) were not performed at wave 5,6,7 and 8.

8) In the introduction, the authors highlight that it can be difficult to distinguish intrinsic capacity from environmental adaptation. However, some of the measures included in the total intrinsic capacity score may capture environmental adaptations. For example, questions on hearing status take account of hearing aid use and questions on sleep disturbance do not take account of environmental adaptations that may have been made to account for underlying sleep difficulties.

Response: We agree with the reviewer. In self-report measures, separating the effect of environment from underlying change is often very challenging. This is particularly the case for sensory measures (self-report on vision and hearing) used in the study. For example, if we are asking older people, “how good is your eyesight for seeing things at a distance, like recognizing a friend across the street”, distance here depends on the breath of the street (10, 50,or 100m), which might vary depending on geographical location of the residence. This issue applies to hearing questions as well. Moreover, both sensory questions take account of assistive devices We have added this limitation in the discussion section.

9) It is reported on page 5, that 6238 participants aged 60+ were included in ELSA wave 4. However, only 2532 were included in the analyses presented. This large drop in N is of concern not only because it is likely to have introduced bias but also because it highlights practical problems in deriving the intrinsic capacity score. Please could the authors clarify why such a large drop in N occurred and comment on similarities and differences in the characteristics of the people they were able to include in their analytical sample and those they had to exclude and the implications of this. The authors should consider methods that may allow them to minimize loss of participants from the derivation of the score.

Response: We aimed to test this concept in relative robust group of older people and therefore excluded those who had already lost IADLs or ADLs. This does not introduce bias since the participants represent the entire study sample without these functional losses at baseline. We have reworded the details provided in the first paragraph of the result section.

10) As it is already well established that all of the variables included in the intrinsic capacity score are related to subsequent risk of morbidity, institutionalisation and premature mortality it is perhaps not surprising that the total capacity score was found to be a powerful predictor of incident care

dependence. I think the authors therefore need to make a much clearer case for what their study really adds.

Response: While not surprising we are unaware of any empirical analysis to confirm this assumption. We have used Bifactor model and path analysis which are advanced statistical approaches to validate the theoretical concepts and predictive validity underpin the Healthy Ageing model proposed by WHO.

11) To be confident that this construct is readily applicable to research and clinical practice (as suggested on page 8), evidence would be required of how it performs in other general community-dwelling and clinical populations. In addition, before it could be applied to 'assist research into the early determinants of functional change' it would be necessary to understand how the score performs in younger populations given only people aged 60+ were included in these analyses. Do the authors have plans for this?

Response: Thank you for this comment. We agree with the reviewer. We have indicated this point in the discussion on implications for future research by emphasizing "If confirmed by future studies, "

12) Time for assessment in both clinical and research settings is limited. A number of different measures are included in the total intrinsic capacity score and many of these are highly correlated. How confident are the authors that each of these measures provides added value? Before routine assessment of such a measure and all its many components could be considered, evidence would be needed that each component was required.

Response: We agree, but our purpose is to frame the construct not to suggest the measures that might be used to measure it. We have extensive other work underway to identify a parsimonious set of measures that might be used in clinical and research settings.

13) Related to point 12, capturing data on biomarkers tends to be more invasive, time consuming and expensive. In addition, there is limited evidence that measures such as telomere length do add predictive value for future outcomes if included in models with measures that capture function at the individual level (see for example, von Zglinicki BMJ 2012;344:e1727 and Martin-Ruiz et al Mech Ageing Dev 2011;132:496-502). It would therefore be helpful to see the authors comment on whether the inclusion of blood-based biomarkers is likely to add value and be necessary especially in light of its associated costs.

Response: Again, our purpose is to frame the construct not to suggest the measures that might be used to measure it. The fact that the data that was available clearly suggest a sub factor which we have labelled Vitality, and that probably lies at a different level from the more overt manifestations of capacity is useful in building this frame. We have tried to emphasise this in the introduction and discussion.

14) A number of other composite scores have been developed that aim to capture different aspects of health and function in older people and which are proposed as tools for monitoring patients and, screening populations to identify those who are at high risk of poor outcomes in later life. For example the frailty index, which is now being applied in clinical settings (see for example, Clegg et al Age Ageing 2016;45:353-60). While these have drawn on different conceptual frameworks, to what extent do they really differ from the intrinsic capacity score especially in terms of their clinical utility?

Response: As Clegg himself emphasizes, frailty is an especially problematic expression of population ageing. There remains considerable disagreement on exactly how to define it and while frailty probably captures some aspects of what might be considered intrinsic capacity, the concepts are very

different. The lack of consensus in fields such as frailty is one of the reasons we saw the need to create a broader construct of health in older age and one of the reasons our work is so important. The World report framed how trajectories of health in older age might be broadly considered using the concepts of intrinsic capacity and functional ability. Moreover, these concepts would be consistent across the whole life course, not merely towards the end of life when frailty normally occurs. This paper is the first step towards describing how one of those concepts may be structured. We then anticipate that researchers in fields such as frailty might reassess their work to consider if this new perspective helps resolve the thorny questions they have not been able to address. The reviewer's comment, in fact emphasises the need for a new way of thinking about these issues.

15) Page 9, it is not clear how the longitudinal nature of the study allowed the authors to examine the direction of causality. In addition, the fact that some of the measures used were self-reported (e.g. for sensory function and sleep) does need to be acknowledged and the implications considered.

Response: As stated above, the limitations of self-reported measures are now expanded on in the limitations section. The longitudinal analysis excluded all older adults with the outcome of interest from the baseline analysis. The analysis then sought to identify the relationship of baseline characteristics with incident outcomes while holding all other information constant. While this cannot categorically confirm causation, it is very suggestive of a causal relationship and is the basis of any longitudinal research.

16) Please consider citing systematic reviews and meta-analyses on the associations between functional measures and survival in place of references 50-52 (e.g. Studenski et al JAMA 2011;305:50-58; Cooper et al BMJ 2010;341:c4467; Calvin et al Int J Epidemiol 2011;40:626-44)

Response: These references have been included, except Calvin et al which is about the relationship between intelligence in youth and mortality, which is not a primary interest of this study.

17) Caution is required in promoting the use of a 'holistic outcome measure' to 'assist research into the early determinants of functional change' especially as evidence suggests that different functional measures have different underlying aetiologies whereby important lifetime risk factors specific to one component may not be identified if only the total score is considered in analyses.

Response: This is why we sought to identify the sub domains of the holistic measure. The problem is that in the past these have been considered outside a broader holistic construct.

Minor points

1) On page 2, when describing the chair-stand test, the authors refer to the eligibility criteria for the balance test. Please correct this.

Response: We have checked the text and made the relevant changes.

2) On page 3, it would be more accurate to report that a trained nurse collected blood samples which were then analysed to generate biomarker data (rather than reporting that trained nurses collected biomarker data).

Response: We have made the changes. Please check page 3.

3) In listing the measures used in analyses please make it clear which are considered for inclusion in the intrinsic capacity score and which are factors that have been used in subsequent analyses.

Response: All measures included in the confirmatory analysis were used to create the intrinsic capacity score and subsequent analyses. The details are described in the statistical section. Please check page 6.

4) Wherever possible please include results tables in the main paper rather than as supplementary data. For example, if BMJ article requirements allow it, it would be helpful for supplementary table s1 to be included as table 1 of the main paper.

Response: We have include results of the important analysis in the main paper and rely on the editors advice as to how much of this information can be included in the paper and how much should be supplementary

5) On page 7, when referring to the 'direct effect' it is important to make it clear that this is the effect that is not explained by multimorbidity i.e. it is possible that this is mediated by other factors not considered.

Response: The issue of residual confounding is now discussed in the discussion section.

6) Please consider adding footnotes to the figures to help readers understand what is shown without having to refer to the text.

Response: We have included the footnote below the figures.

7) In table 1s it would be helpful to see the distributions of the variables that are included in intrinsic capacity score and for this information to be stratified by sex (given there are sex differences in the distributions of many of these variables).

Reviewer 3

Recommendation:

Comments:

The only fault that I had, was whether the manuscript was so detailed in the description of the statistical methodology; the exposition of the results; and the subsequent pathway analysis to care dependence; that some of the more technical aspects might be better incorporated within a web based annex, with the principal findings streamlined for publication in the journal itself. Finally, knowing the competition for such papers in the BMJ, if the manuscript does not meet the publication threshold, I wonder whether an editorial, or an abbreviated review of the topic, would be appropriate for the readership. I have little doubt that this conceptual framework will dominate the future of comprehensive geriatric assessment among healthcare systems worldwide, and has the great advantages of doing so regardless of underlying population wealth.

Response: Thank you for the comment. The concept of intrinsic capacity is new, therefore, it requires sufficient statistical analysis and information to justify the approach and also guide future research. Regarding online, we received mixed feedback from the reviewers. Some reviewers indicated that we should retrain all the tables from the online annex in the main paper, while other suggested moving additional information to annex. We have considered the comments and put what we consider the most important information in the main papers and additional information in the annex. But we are very open to the editor's advice.

Reviewer: 4

Care dependence was assessed using self-reported limitations in the Basic Activities of Daily Living (ADL) and Instrumental Activities of Daily Living (IADL). The description suggests that the authors used scores based on no/yes (0/1) answer only, with no consideration for level of difficulty. Why not?

Response: Care dependence measures (ADL and IADL), used in ELSA, assessed whether respondents required help (from other person) to carry out the activities. Our interest is to understand whether need for help increases with high or low baseline intrinsic capacity scores, not severity of ADL or IADL difficulties.

The multimorbidity score was based on information on doctor diagnosis of 14 conditions, some of which are more interrelated than others, as the cardiovascular ones. On the other hand, they just include cancer as a broad category. Was the choice of the 14 conditions based on what was available in the ELSA cohort or it was based on a pre-selection of conditions that matter. What the authors recommend as a good repertoire of medical conditions to add to this score. The list and number of conditions included will influence categorization, especially because the multimorbidity variable is grouped as none, 1 or 2, and 3 or more conditions.

Response:

This is a valid point but beyond the scope of our paper. We used the data available to us and are not recommending the same methods for other research. In fact the issue of comorbidity is complex and warrants a more comprehensive analysis in future research.

The authors did not mention, but I presume that the presence of physical defects, including upper or lower limb loss or impairment, poor manual dexterity, and damage to one or multiple organs of the body should be considered either separately or as part of the multimorbidity score.

Response: The reviewer is correct, however that was not the purpose of our research and beyond the scope of our analysis. We have tried to build a construct that is framed from a functional perspective as distinct from a disease based model. We included multimorbidity in our model, not to propose a quantitative model where disease and capacity could be added to give a predictive score, but to check that intrinsic capacity captured information that was missing from the assessment of multimorbidity and that it added prognostic value beyond a more traditional disease based assessment. This proved to be the case and in fact intrinsic capacity had a greater influence on outcomes than multimorbidity. However, the multimorbidity score was cursory and more advanced work has been done by other researchers to better capture and describe these characteristics.

In the analysis, the authors state that "No imputation was performed in the analysis and participants with missing data were excluded, leaving a study sample of 2532 with complete data for the EFA and CFA analysis". Considering the initial sample 6238 individuals at Wave 4, it means that only 40,6% of the sample was included. Thus, it cannot be discarded some differential losses related to physical and mental capacity items that may impact the overall analysis. Table 1 should present the distribution of characteristics of the individuals that contributed to the analysis (n=2532) as well and p-value to indicate differences with the total sample (n=6238).

Response:

We have redrafted Table 1 to be more clear. However, we disagree with the need to compare the study population with the total sample. By definition, those not included will be different since they have already lost activities of daily living.

Finally, I think the authors make a very good point when they say “...the combined score takes no account of thresholds that may exist within each subfactor. For example, cognitive capacity may fall to the point where it becomes impossible for an individual to survive without appropriate care and support, even though they may retain perfect capacity in each other domain and thus retain a relatively high total capacity score.” I think the authors could develop this a bit further and perhaps indicate that the use of a multidimensional approach would not be adequate (or necessary) when a clearly incapable health condition (physically or mentally) exists.

Response: Good point and we have expanded in the discussion

Reviewer: 5

This is an interesting paper in an area of great and increasing importance. The paper is well written but this reviewer has some concerns about the methodology particularly regarding the collection of the baseline data. These are concerns detailed below. This reviewer recommends an independent statistical review of the paper as the statistical methodology is complex and rather specialized.

Major Concerns

1. Paragraph 2; line 4: All subjects ‘signed fully informed consent’. Though this is clearly appropriate it does censor the study data particularly as one of the major covariates in the comorbidity analysis was dementia. The authors should comment on this in the discussion section as an important weakness of the study (particularly given that dementia was such a predictive covariate).

Response: Discussion amended to include (although participants with significant dementia are probably excluded from the study sample since no participants had ADL or IADL loss at baseline).

2. Page 3; lines 5-9: This reviewer is concerned about the apparent lack of understanding of lung function measures. It appears that the measure that the authors have conducted is forced expiratory volume in one second (my underlining) and not forced expiratory volume (line 5). Further to this they (line 8) state that they have conducted forced expiratory volume in one minute which is clearly incorrect. Equally importantly they (line 8) state that they have used the ‘highest technically satisfactory measure’ of forced expiratory volume. They do not state which criteria or guideline they base this latter assertion upon. Forced expiratory volume in one second has very clear international acceptable guidelines for its assessment. It is unclear to this reviewer whether the authors are aware of these and whether they following them.

Response: Thanks for picking this up. We have revised the text. Criteria for unsatisfactory attempt is provided. Please check page4.

3. Page 3; lines 20-27: I am very concerned by this paragraph. Hearing impairments in particular are usually under-reported by older people (and probably by younger people as well). What validation was there for such self-reported measures of hearing? I am less concerned about the self-reported measures of vision. However, once again the authors are working from data that is available to them and can do no more - but (also once again) they should cite this as limitation of the study in the discussion section.

Response: We agree and have responded to this limitation in the discussion section. We strongly discourage using self-reported information for sensory capacity in future research. We were limited by the nature of this secondary analysis and do not encourage subjective assessments of these capacities. But as discussed under reviewer 2, we are not proposing a measurement approach but a conceptual frame and we feel the data that is available in ELSA is adequate for that purpose.

4. Page 3; lines 32-35: The authors need to provide a reference for their assertions regarding taking animal naming as a measure of executive functioning and taking letter cancellation as a measure of processing speed.

Response: We have included the references. Please check page 5.

5. Page 4; line 7: Why were myocardial infarction and angina separate measures of multimorbidity? They represent the same disease. Later on in the paper (eg Figure 3) these and another comorbidity in the list quoted at the top of page 2 (congestive heart failure) seem to be lumped together as 'heart disease'. The authors need to be clear on whether each of these covariates (myocardial infarction, congestive heart failure and angina) have been analysed separately or individually. If the latter they need to justify the separate analyses for myocardial infarction and angina.

Response: The data set used in the study is a secondary dataset. Most diseases classification are constructed variables. Where ever possible, scores were analyzed separately and presented for individual conditions. We have clarified this and some of the other reviewer's comments in the methods section

6. Page 5: The authors do not define the abbreviations EFA and CFA at any point.

Response: Abbreviations are provided in the statistical section. Please check page 5.

7. Page 7; first full paragraph: This is the beginning 'so what' section of the paper, ie what is the usage of intrinsic capacity? As such I do not believe that Table 5s and Figure 1s should be supplementary to the paper. They should be part of the main published article.

Response: We agree. The tables are moved to the main paper.

8. The Discussion Section is particularly powerful (most especially the third, fourth and fifth paragraphs on Page 8). However, (page 9; lines 6-8) whilst technically accurate needs to take account of my comments regarding the potential flaws in the methodology as mentioned above.

Response: The discussion has been amended to take account of these concerns.

9. Figures 1, 4 and 5: A key is needed explaining what the numbers adjacent to the lines within the figures mean.

Response: We have included a footnote for interpretation of number the direction of path.

Minor Points

1. Page 1; line 22: I would suggest that the words 'can sometimes' are deleted.

Response: Words " can sometimes" are deleted.

2. Page 2; line 55: I assume that when the authors refer to 'average' are they in fact referring to 'mean'? If so, they should say so.

Response: Mean is now mentioned in the parenthesis.

3. Page 3; line 17: Note the inaccurate spelling of the word haemoglobin.

Response: We have changed to British spelling.

4. Page 6; paragraph headed 'construct validity': I would suggest that this paragraph is moved above the paragraph headed 'reliability of the factor scores'

Response: We would like to retain the same order. Before testing the construct validity, it is important to understand the reliability of the score. However, we are happy to be guided by the editor.

5. Page 6; line 32: By 'these conditions' this reviewer presumes the authors mean the comorbidities. This should be stated specifically as it is not stated in the bolded paragraph heading.

Response: We have revised to text for clarity. Please check page 7, last paragraph.

VERSION 2 – REVIEW

REVIEWER	JEAN YVES REGINSTER University of LIEGE, BELGIUM
REVIEW RETURNED	11-Jan-2019

GENERAL COMMENTS	This paper is interesting, timely and of potential major importance to improve the life of elderly subjects The reviewers comments are appropriately taken into account
--

REVIEWER	j philip miller Washington University School of Medicine USA
REVIEW RETURNED	10-Feb-2019

GENERAL COMMENTS	Very dense, technical description. Limitation about completeness of both variables and subjects mentioned, but not emphasizes as much as desirable.
---

VERSION 2 – AUTHOR RESPONSE

REVIEWER 2 COMMENT: Limitation about completeness of both variables and subjects mentioned, but not emphasizes as much as desirable. ADDED COMMENT INCLUDED PAGE 2 AND PAGE 11